# Deep-Learning-Based Character Recognition from Handwriting Motion Data Captured Using IMU and Force Sensors

**DOI:** 10.3390/s22207840

**Published:** 2022-10-15

**Authors:** Tsige Tadesse Alemayoh, Masaaki Shintani, Jae Hoon Lee, Shingo Okamoto

**Affiliations:** Department of Mechanical Engineering, Graduate School of Science and Engineering, Ehime University, Bunkyo-cho 3, Matsuyama 790-8577, Japan

**Keywords:** smart pen, handwritten character recognition, deep learning, inertial sensor, force sensor

## Abstract

Digitizing handwriting is mostly performed using either image-based methods, such as optical character recognition, or utilizing two or more devices, such as a special stylus and a smart pad. The high-cost nature of this approach necessitates a cheaper and standalone smart pen. Therefore, in this paper, a deep-learning-based compact smart digital pen that recognizes 36 alphanumeric characters was developed. Unlike common methods, which employ only inertial data, handwriting recognition is achieved from hand motion data captured using an inertial force sensor. The developed prototype smart pen comprises an ordinary ballpoint ink chamber, three force sensors, a six-channel inertial sensor, a microcomputer, and a plastic barrel structure. Handwritten data of the characters were recorded from six volunteers. After the data was properly trimmed and restructured, it was used to train four neural networks using deep-learning methods. These included Vision transformer (ViT), DNN (deep neural network), CNN (convolutional neural network), and LSTM (long short-term memory). The ViT network outperformed the others to achieve a validation accuracy of 99.05%. The trained model was further validated in real-time where it showed promising performance. These results will be used as a foundation to extend this investigation to include more characters and subjects.

## 1. Introduction

The recent growth in miniaturization technology in electronic components has contributed greatly to the development of convenient human-computer interaction devices. The size, weight, and portability of these devices have been the main focus of the human-computer interaction research community. Most existing technologies rely on touch panel screens equipped with an intuitive graphical user interface. With the advancement in computing devices, pattern recognition methods, and human-computer-interface technologies, the field of handwriting recognition has become popular. Low-cost sensor technology devices, particularly motion sensors, and interactive technologies are being rapidly developed and used in gesture recognition, activity recognition, motion tracking, and handwriting recognition applications [1,2,3,4,5,6]. For people in educational institutions, such as students, notetaking and typing occupy much of their daily life.

Compared to conventional input methods using keyboards and touchscreens, handwriting character recognition using inertial sensors is an emerging technique. As a result, many traditional systems are being digitized at a fast pace. One of them, which has recently received attention, is the automatic digitization of handwritten characters. Digital notetaking systems are changing the way we take notes in classrooms or write memos during a meeting and in other contexts.

Handwriting character recognition methods mainly fall into two broad categories: online and offline recognition methods [1,2]. An offline recognition method performs scanning over previously written static images of text files. It has been a popular recognition method in various fields, including banking [3], the healthcare and legal industries [4], and postal services [5]. One popular offline method is optical character recognition (OCR). It has been widely used to digitize texts from old manuscript images. To this day, it has been the dominant approach for offline text extraction from image documents. The core technique of OCR is image processing, which comes at the cost of computation time. Hence, OCR is not optimal for recording real-time handwriting. Moreover, it is more effective with keyboard-typed texts than handwritten texts. On the other hand, online handwriting recognition commonly employs real-time time-series input data characterizing the spatio-temporal features of the characters [7]. Touch-based online recognition methods are also common in smart devices, such as tablets and mobile phones, where end-users digitize their handwritten texts using stylus pens or finger-tip touches on screens. These systems utilize precise positioning of the tip of the input stylus or finger to trace the point of contact. Hence, such systems are widely used in notetaking in everyday life. Some prominent notetaking mechanisms are the use of stylus pens with tablets, e-ink devices, and smart pads. However, these approaches do not enable the conversion of handwritten characters into machine-readable format, rather handwritten notes are stored as images or portable document files. In addition, these methods require a special smart surface/pad and special applications for precise tracking of the stylus tip position. Hence, their market price is quite high, which inhibits users from using ordinary white paper. The fact that handwriting is a huge part of our daily lives, particularly for people such as academics and office workers, necessitates a cheaper and more compact digital writing approach.

The concept of using an inertial sensor-based digital pen as an online and non-touch handwriting recognition method has been investigated for the last decade [8,9,10,11,12,13]. Wehbi et al. employed two inertial sensors at both ends of the pen to capture the spatial motion of the pen. In addition, they attached a force sensor to the back of the ink chamber used to check whether the pen tip was in contact with the writing surface or not. Their data analysis was performed with an end-to-end neural network model called CLDNN, which is a combination of convolutional neural network (CNN), long short-term memory (LSTM) and fully connected (FC) [8]. Antonio et al. developed a digital pen equipped with 9-axis inertial measurement unit (IMU) data [9]. Wang et al. [10], Srinivas et.al., and Patil et.al. developed digital hardware to collect character data written in free space without any size limitation [11]. Patil et.al. utilized a dynamic time warping (DTW) algorithm for real-time handwriting recognition, while Wang et.al. estimated the position of the pen through inertial data integration. Similarly, the authors of [12], used the generative adversarial network (GAN) to classify in-air written words, where the training data was collected by employing a built-in inertial smartphone sensor. Moreover, [13] used an unsupervised network called the self-organizing map, to recognize the 36 alpha-numerals. However, the use of only inertial/motion data for classifying notetaking-level, small-sized characters is difficult as the IMU data are quite similar due to the limited range of motion. Hence, an additional low-level sensor is needed to boost the performance of such systems.

Some investigations have supplemented inertial data with additional low-level sensor data. A study by [14], added strain gauges to inertial sensor data. The authors claimed that the strain gauge data helped in compensating for the intrinsic inertial sensor drifting error. Their algorithms were evaluated by experiments conducted using a robotic manipulator. Another study by Schrapel et al. added a microphone in addition to an IMU sensor. The microphone picked the stroke sounds, which the authors assumed had complementary properties with the motion of the tip. The authors claimed to have improved the performance of their neural network-based classification system [15]. Another non-touch and non-vision-based sensing method used for mid-air handwriting is radar [16]. In [17], three ultrawideband radar sensors were employed to track the trajectory of a hand in mid-air for digit writing. This system required a proper setup and multiple devices, which increased the cost of the whole system.

Others have used a special camera sensor to track the tip of the input device [18]. Tsuchida et al. utilized a leap motion controller, which was equipped with optical cameras, to detect handwritten characters in the air. They claimed to have successfully classified 46 Japanese hiragana characters and 26 English alphabet letters [19]. Furthermore, Hsieh et al. also utilized a trajectory dataset calculated from 2D camera data to train a CNN model for the recognition of in-air-writing [20]. The shortcoming of air-writing is that, despite its good performance for large-sized characters written on a free space, it cannot be easily implemented for notetaking-size character recognition. In addition, cameras are prone to obstruction and require environmental lighting to work properly.

Another stage in handwritten character recognition was the development of an algorithm to correctly identify each character by reconstructing the pen trajectory by classifying collected data. Attitude computations, filtering, and inertial data integration were employed in [10,13] to reconstruct the trajectory of the experimental device. The authors of [11] used dynamic time warping to distinguish characters based on their temporal data similarity. However, due to their wide range of application, from finance to construction, from information technology to healthcare application, machine learning methods are becoming increasingly prominent in handwritten character recognition [21,22,23,24,25]. In [12], a GAN-based machine learning method was used to solve a classification problem, while [8] describes the use of a deep-learning-trained neural network method to collect sufficient data.

The challenge with using inertial sensors for a digital pen is that a significant distinction cannot be made between the data for different characters. If an inertial sensor is used to write characters that are similar in size to normal note characters, the resultant data show little difference for all the characters. In addition, inertial sensors suffer from intrinsic noises and drifting errors. The studies described took advantage of writing characters in free space so that the classification problem would be easier. The other alternative is vision/camera-based handwriting character recognition. However, such techniques can suffer from occlusion, high computational costs, and sensitivity to lighting conditions. Hence, none of these methods can be implemented effectively for notetaking in real-time.

In this paper, a smart digital pen was developed for online handwriting recognition. There are two vital pieces of information that need to be obtained from a pen. These are the direction of motion and the pen tip’s trajectory data. To satisfy these requirements, we developed digital pen hardware equipped with an IMU sensor and three tiny force sensors. The trajectory information was mainly obtained from the inertial sensor while the force sensors supplemented the system with the direction and trajectory information of the pen. The developed digital pen was slightly thicker than an ordinary ballpoint pen. The IMU sensor and force sensors were carefully placed in positions that enabled them to record relevant and higher amplitude motion and force data. All experiments were undertaken on normal notebook paper with normal notebook character size. A neural network trained by a deep learning method was chosen as conventional methods suffer from drifting errors. Four neural network models, including CNN, LSTM, deep neural network (DNN), and vision transformer (ViT) were investigated for the classification of 36 handwritten English alphanumeric characters. These included the numerals from ‘0’ to ‘9’ and the 26 lower-case English alphabet letters.

The main contributions of this paper are:
(1)Development of handwriting recognition digital pen hardware, slightly thicker than a typical ballpoint pen, that can be used anywhere, without any external reference device or writing surfaces.(2)Development of a deep-learning algorithm that combines inertial and force data of a pen which was successfully tested for typical notebook-sized alphanumeric characters.


## 2. Hardware Components

In this section, the hardware components of the developed digital pen will be introduced in detail. The digital pen used in this study was developed in our previous research [26]. The hardware architecture of the pen is shown in Figure 1. The electrical circuit board was designed in our RoBInS (Robotics and Intelligent Systems) laboratory and outsourced for printing. The 3D plastic pen body design and printing and the assembly of all the hardware were also undertaken in our laboratory. Below are the materials used in this study.

IMU sensor: A six-channel (three linear acceleration and three angular velocities) IMU sensor was utilized to measure the motion of the pen. An LSM9DS1 IMU sensor, a small chip embedded in the Arduino Nano 33 BLE board, was utilized in this research. The specifications for this sensor are shown in Table 1. The placement of the IMU sensor is decisive. It is preferable to mount the sensor on parts of the pen that manifest larger motion. Two likely locations are closer to the pen tip and around the tail of the pen at the back. By comparing the magnitude of the pen’s motion at both of its ends during handwriting, a bigger motion at the tail side can be observed. This is because a higher moment is produced on the back of the pen than on its front side during normal handwriting. Motion data with a bigger magnitude helps neural network models extract more features and to speed up the discrimination capability of the network. Hence, the Arduino microcomputer (with embedded IMU) was mounted at the tail of the pen.

Force sensors: One way to capture the direction and length of the digital pen tip’s motion is to install force sensors close to the pen tip. To record the magnitude of the force exerted on the pen tip in any direction, it was necessary to enclose the ink chamber of the ballpoint pen with force sensors. Three tiny force sensors were placed around the ink chamber at 120° to each other. Physically, the pen should not be bulky, but instead be easy to handle and convenient for writing. Therefore, the force sensors were chosen to be as small as possible. Alps Alpine (an electric company headquartered in Tokyo, Japan) HSFPAR303A force sensors, shown in Figure 2 were chosen for this study. Their small size made them possible to be accommodated by the designed pen body part.

As can be seen from Table 2, the force sensor’s output for a 1 [N] force difference was 3.7 [mV] which is quite small. Hence, an amplifier circuit was designed and added to its output, as can be seen in Figure 2. To accommodate both the force sensor and the amplifier circuit, a small printable circuit board (PCB) was designed using KiCAD software. The designed circuit was outsourced for production. The newly produced force sensor had dimensions of 28 × 4.57 × 2.06 [mm].

Microcomputer: As mentioned in the previous section, the core computing device inside the pen hardware was an Arduino Nano 33 BLE microcomputer. It was equipped with an IMU sensor which enabled it to measure the pen movement from the accelerometer and gyroscope sensors. The microcomputer is in charge of collecting and arranging the inertial and force sensor data of the pen and later transmitting all the data to a computer through a serial communication channel. The Arduino microcomputer has an onboard ADC chip to convert analog force sensor data into digital form.

Pen body: The pen body was designed on 3D CAD software, SOLIDWORKS, to accommodate all the hardware parts. It was later printed on a 3D printing machine from an acrylonitrile butadiene styrene (ABS) thermoplastic material which can be seen in Figure 1.

Computer: A LB-S211SR-N mouse laptop computer, manufactured by Mouse Computer Co., Ltd., a company based in Tokyo, Japan was used for storing character datasets and running trained neural network models during real-time inferencing.

The schematic diagram of the whole system setup is shown in Figure 3, where the digital pen and computer components and the process are depicted.

## 3. Data Collection and Preparation

In this section, we introduce the data collection process and explain the preprocessing performed on the data that was used to train our model. The data of the handwritten alphanumeric characters were collected from six male right-handed volunteers at 154 [Hz] sampling frequency. A particular instruction was not given to subjects on how to write characters, rather they followed their natural way of writing. The writing speed for these alphanumeric characters was chosen freely by the subjects. To avoid the difficulty of labeling the dataset later, each subject was asked to write one character 50 times. This made up one set of data. In between each subsequent character, a brief pause was taken. This brief pause was useful for trimming the raw data to small sizes of datasets before training. Each time a subject completed writing a character 50 times, the data was saved in the computer, and the subject started writing the next alphanumeric character. In this way confusion during labeling would not occur as each file could be saved with a name that indicated the subject participated and the alphanumeric character that he/she wrote. Enough pause was taken between each consecutive character to make the dataset segmentation work more easily in the later stages of training preparation.

The collected time-series data of each character was later segmented into smaller time-series data blocks that contained only a single character’s data. This segmented single character data was considered as one dataset during the neural network training. Each of these datasets was extracted from its corresponding set of data, which contained the same character written 50 times in succession. Hence, to segment the dataset from the set data, a shifting window, with a fixed time length of 1.3 s (200 data samples wide), was applied to each set. All the characters in this study were written in less than 1.3 s; hence, this range was chosen to include all the characters.

Before starting the segmentation process, it was necessary to distinguish three pen events: the pen is not grasped by the subject hence not writing; the pen is held by the fingers of the subject but not in writing mode; the pen is currently held and is writing. To differentiate these three events, the measured force data is useful. When the pen is on the table and not touched, the force measurement is at its lowest value. However, when writing starts, the force reading goes up the moment the pen is grasped with the fingers and stays almost constant unless writing starts. While writing, the reading of the sensors continues to increase. Thus, by identifying a proper threshold force value, we can distinguish among these events. Before commencing data collection, each subject held the pen for brief seconds without doing anything. Hence, the average value of the force reading for the first brief seconds was taken as the dividing line between writing events and not-writing events. The timestamp at which a reading was 100 [N] above or below this value was deemed to be the beginning of a character’s handwriting data. This helped us to identify the beginning and ending timestamps of the alphanumeric character data. The beginning timestamp was advanced by a few samples (20 samples) to make sure the shifting window covered the whole character data. Figure 4 shows a diagram of the segmentation process performed over the collected sample data. The segmentation was made possible with the help of the force data and a brief pause used during the data collection. The difference in some characters’ force measurements could even be distinguished by the naked eye. As an example, the datasets of six characters are shown in Figure 5.

After trimming the raw data, each dataset has a shape of 9 × 200, where the row represents the 9-axis inertial and force measurements (3-axis accelerations, 3-axis gyroscope, and 3-force sensors), while the column is the width of the shifting window. As a preprocessing step, the force sensor data and the IMU sensor data were combined and restructured before feeding the data to the neural network, as shown in Figure 6a. The restructuring was introduced to increase the extraction of spatial and temporal correlation features during model training. Here, the data structuring method proposed in our previous research [6] was utilized for deeper feature extraction in handwriting motion. The final shape of a single dataset was 18 × 200, where the first dimension was a duplicated version of the 9-channel multivariate data (force, acceleration, and angular rate, each with three channels), and the second dimension was the shifting window width. These datasets can be treated as virtual images when used as input for the CNN and ViT networks. An example is shown in Figure 6b.

A total of 10,800 datasets were prepared, 300 for each alphanumeric character. Out of these, 8330 datasets were used for neural network training, 1470 datasets were used for validation during the training, and the other 1000 datasets were used for testing the trained neural network model.

## 4. Structure of the Neural Networks

In this section, the architecture of the neural networks used for training and related procedures is presented. Four neural network architectures were proposed to develop an end-to-end handwriting recognition model.

The first is an LSTM network. Recurrent neural networks (RNNs), particularly LSTM networks, are common in handwriting and speech recognition applications because of their ability to transcribe data into sequences of characters or words while preserving sequential information. As the dataset in this study comprised sequential time series data, LSTM was a good candidate for deep-learning training. Since each dataset had a temporal axis length of 200 samples, the input layer of the LSTM network had a 200-time steps data input layer. The whole network included a single LSTM layer, which had 100 units, followed by three fully connected layers. The output of the last LSTM unit was then fed into the two fully connected layers which had 256 and 128 node sizes, respectively. Finally, a 36-class softmax classifier was added.

The second alternative network was the CNN model. CNN is among the most popular neural networks owing to its excellent performance in image processing and pattern recognition. As mentioned in the previous section, the input datasets were treated as virtual image inputs to the CNN. To reduce complexity, a two convolutional layer CNN was prepared. In the first convolutional layer, the 18 × 200 input data was convolved by 256 convolutional filters of size 2 × 2 to extract the spatial and temporal features. This was followed by a 2 × 3 downsampling pooling layer. Similarly, the second convolutional layer filtered the output array from the first layer with 128 convolutional filters of size 3 × 3. Again, to reduce the larger temporal axis dimension, downsampling using a 1 × 2 max-pooling layer was performed. Next, the result from the second convolution layer was converted into a 1D vector of size 512. Lastly, the probability distribution-based classifier called softmax was applied to classify data into 36 alphanumeric characters. The designed CNN model is shown in Figure 7.

As a third alternative, DNN was also investigated in this paper. The models presented in this paper take multivariate time-series input data samples of different lengths, comprised of 18 channels, representing the duplicated tri-axial measurements of the two IMU sensors in addition to the force sensor. The layers of the DNN have a size of 512, 256, and 128 from input to output side order, respectively.

The fourth network candidate was a vision transformer (ViT). A transformer model is a recent and popular network that uses the mechanisms of attention, dynamically weighing the significance of each part of the input data. As a transformer variant, ViT represents an input image as a sequence of smaller image patches (visual tokens) that are used to directly predict the corresponding class labels for the input image [27]. Patches are then linearly projected to a feature space which is later supplemented with the positional embeddings of each patch. These positional embeddings provide the sequence information to the transformer encoder network. Then the encoder calculates a dynamically weighted average over the features depending on their actual values. At the end of the network, a simple feedforward network is added to perform the classification of each input image. The block diagram of the full network is shown in Figure 8.

The original virtual image of each character, which was 18 × 200, was split into 90, 2 × 20 smaller virtual image patches, as shown in Figure 8. Each patch was projected into a 64-dimension feature vector, which was later added to the position embeddings. In the end, a two-layer feedforward layer was added to complete the classification process.

All the models were compiled based on the categorical cross-entropy loss and a learning rate of 0.00001. To improve the performance of the networks, batch normalization and regularization methods, such as dropout and weight regularization, were applied during training. The models were updated using the Adam optimizer algorithm for about 300 epochs, using a minibatch size of 50 datasets. The exponentially decreasing learning rate parameters, mini-batch size, and other hyperparameters were determined after several trial trainings. All hyperparameters were fine-tuned by trial and error for the different neural networks.

The training was conducted in a 16 GB RAM and Core™-i9, XPS 15 Dell, equipped with NVIDIA^®^ GeForce RTX™ 3050Ti GPU. The open-source deep learning API, Keras, was adopted for training. The Python programming language was used for dataset preparation, training, and inferencing. However, the programs for data collection were based on the Arduino C programming language.

## 5. Results and Discussion

In this part of the paper, the training results of the four models will be discussed. Before proceeding with the training, the datasets were properly segmented, structured, and shuffled. Next, the prepared dataset of size 10,800 was divided into its three corresponding dataset categories: training, validation, and testing. All four neural network models were trained with the same training, validation, and testing dataset. Therefore, a fair training performance comparison could be made. The training Python codes of the networks were developed on PyCharm Professional IDE.

The training conditions for the networks were set as follows: An epoch size of 200; Adam with a learning rate of 0.00001; and a categorical cross-entropy loss function. These learning settings were applied to all the networks to make the comparison easier. Figure 9 and Figure 10 show the four networks’ validation losses and validation accuracy, respectively for the four network models. During training, every time the network models were updated, they were tested using the validation datasets. As there was no overlap among the three dataset categories, the validation datasets were unseen datasets for the networks. Hence, the training progress can also be observed from the validation loss graphs. Since the CNN size was large in comparison to the other three, its learning progress was slower at the beginning but caught up with the number of epochs, as depicted in Figure 10.

Another neural network training performance measurement metric is accuracy. Graphically, the corresponding validation accuracy is shown in Figure 10. As can be seen from the figure, DNN performed worse than the others. Even though LSTM was slightly better than DNN, it exhibited some ripples during its training. This could be due to LSTM’s excellence in extracting temporal features but not spatial features. Spatial features can be extracted to some degree using CNN and ViT. In particular, ViT can discriminate the temporal attributes well using their positional embedding information and, to some extent, spatial information through their patches. Hence, ViT outperformed CNN, which was excellent at extracting spatial features.

To indicate the advantage of having an additional force on top of the inertial sensor data, a training comparison was made between only inertial, only force, and both inertial and force sensor data. The result is shown in Table 3. As can be seen from the table, the force sensors improved the result of the ViT network by 1.3%. The advantage of utilizing both data types together over force sensor data only was also investigated in our previous paper [26]. The results showed that a 1.6% performance improvement was achieved by combining both types of data when compared to only inertial-based training results. Hence, in this study, we focused on the combined dataset.

Another unseen dataset, the testing dataset, was used to validate the trained neural network models. A promising score was obtained from all the network models. For the sake of clarity, only the best results from the trained ViT model are shown in Figure 11 as a confusion matrix. The columns and rows represent the predicted testing alphanumeric characters and the ground truth alphanumeric characters, respectively. Looking at the diagonal cells of the diagram, the model achieved an excellent result for the 1000 testing dataset. The diagonal cells represent the correct predictions, while the rest indicate wrongly classified characters. Similar shape characters would have been hard to discriminate using only IMU sensors; however, the force sensors capture more subtle differences, as can be seen in Figure 5. Additionally, other evaluation metrics, such as macro-averaged recall and F_1_ scores, are shown in Table 4.

To the best of our knowledge, there are no public datasets that have been acquired in a similar way to ours. Either the few existing related papers do not provide open data, or they did not collect data for alphanumeric characters at a notebook font size level. Hence, we could not find a relevant study to compare our study with. However, as an extra layer of validation, our trained model was tested further in real-time. The corresponding result is shown pictorially in Figure 12. The word “hello world” was attempted using the trained model in real-time. As can be seen from the figure, the prediction was correct except for the letter “o” which was recognized as the letter “c”. This could have been because both letters “o” and “c” have circular curves.

## 6. Conclusions and Future Work

In this study, we developed a novel digital pen that embodies two main sensors: inertial and force sensors. Handwriting data for the 36 alphanumeric (10 numeral and 26 small Latin) characters were collected from six subjects. The data collected were carefully segmented with a shifting window to prepare the datasets so that they would fit the neural network models during training. The segmented datasets were restructured into an 18 × 200 2D array of virtual images. As a validation method, the dataset was used to train four neural network models (ViT, CNN, LSTM, and DNN) using deep-learning methodologies. ViT performed better than the other three with a validation accuracy of 99.05%. It was also shown that complementing inertial data with force sensor data improved the overall performance of the system. Furthermore, the system was also tested for real-time character prediction, where it showed a promising result.

Even though the datasets were small, this study will provide a basis for more research to deepen the automation digitizing of handwritten characters, especially from handwriting motion. It has also provided a strong foundation for future extension of the study. In the future, this method will be extended to include more subjects and more alphanumeric and special characters. As only right-handed young men were included as participants, we are planning to include different age groups, left-handed people, and people with different backgrounds. Additionally, more dataset structuring methods and new neural network models will be investigated to improve the performance. Ultimately, the goal is to produce a robust real-time predicting system to recognize words from continuous writing.

## 7. Patents

The results of this study partially validate the recently published patent: “Learning System, Inference System, Learning Method, Computer Program, Trained Model and Writing Instruments”. Japanese Patent Application Number P2021-29768.

## Figures and Tables

**Figure 1 sensors-22-07840-f001:**
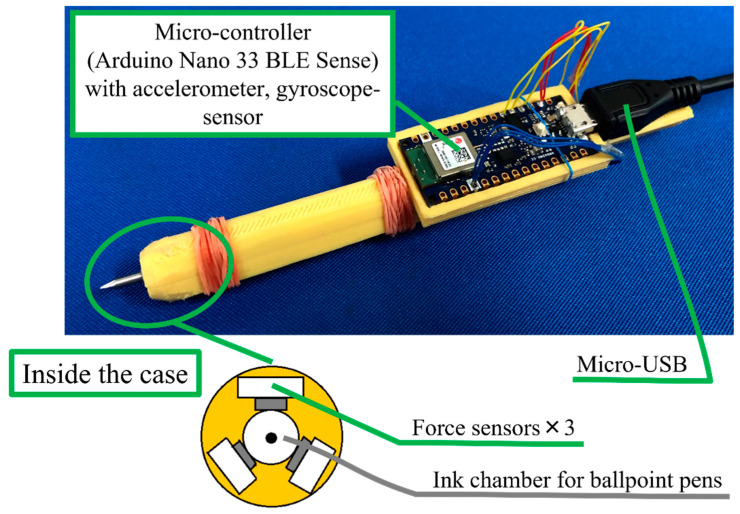
The developed smart pen hardware system.

**Figure 2 sensors-22-07840-f002:**
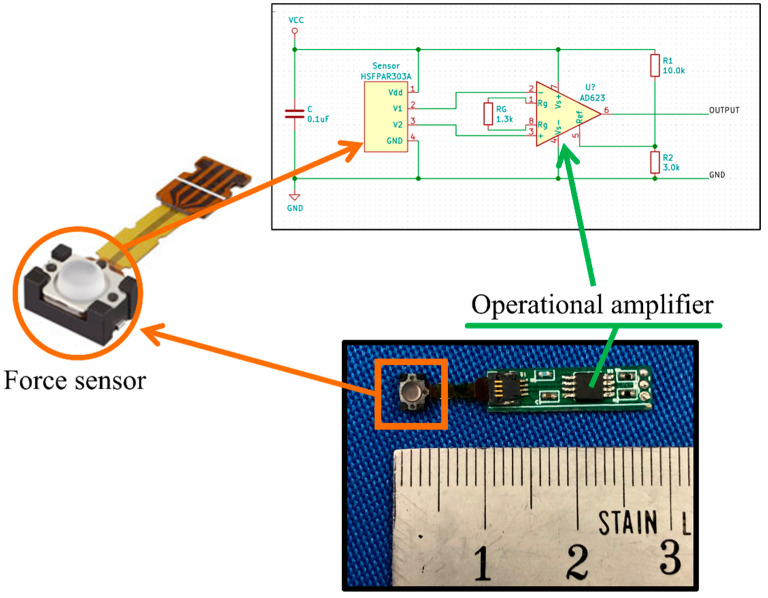
Modified force sensor internal circuitry.

**Figure 3 sensors-22-07840-f003:**
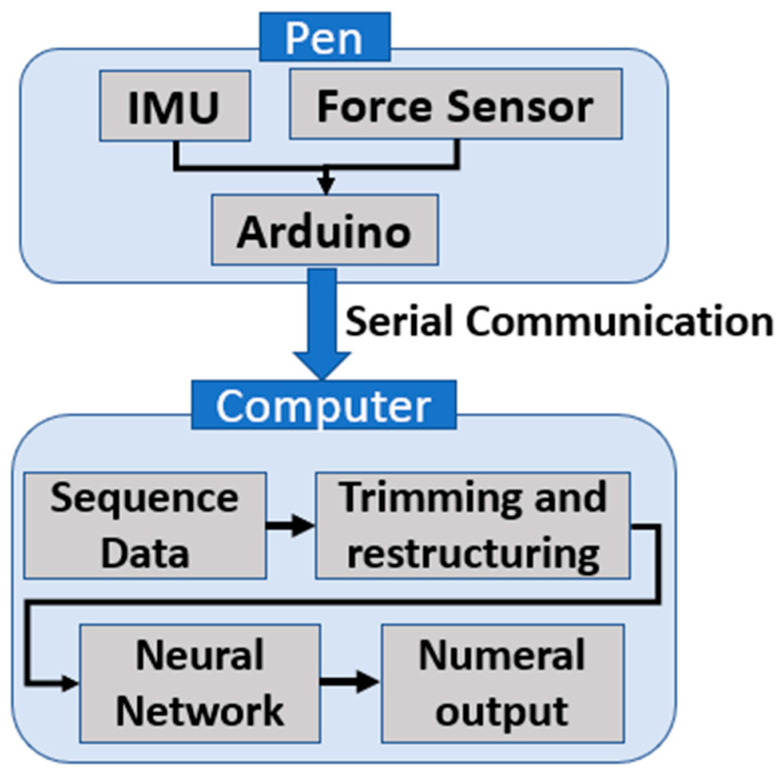
Overall system diagram.

**Figure 4 sensors-22-07840-f004:**
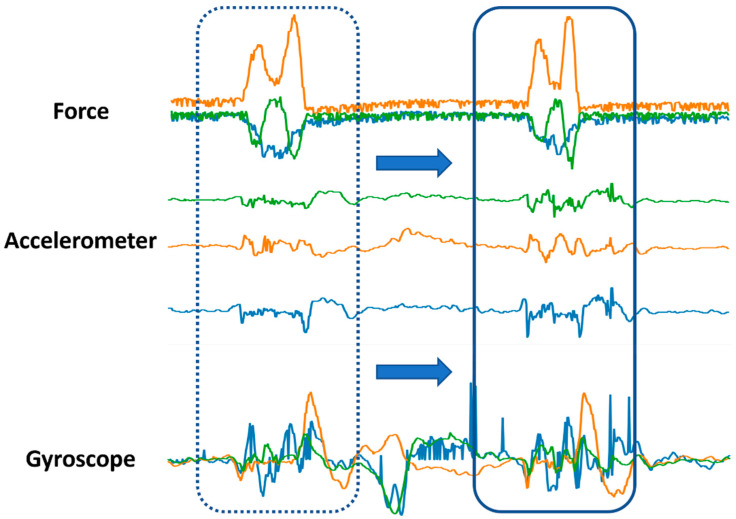
Dataset construction through segmentation. The units are [N], [g], and [deg/s] for the force, accelerometer, and gyroscope, respectively.

**Figure 5 sensors-22-07840-f005:**
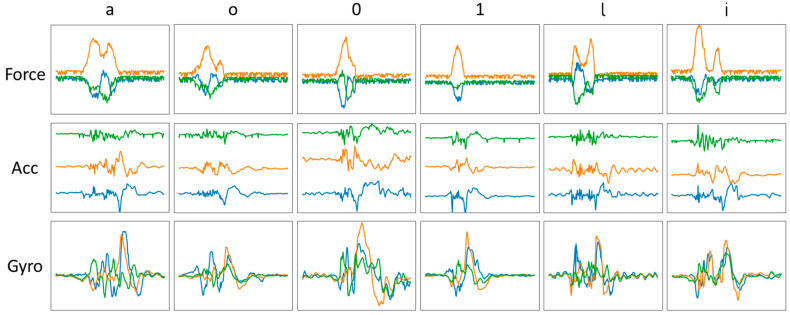
Raw dataset of lower-case letters: ‘a’, ‘o’, ‘l’, ‘i’ and numbers: ‘0’ and ‘1’.

**Figure 6 sensors-22-07840-f006:**
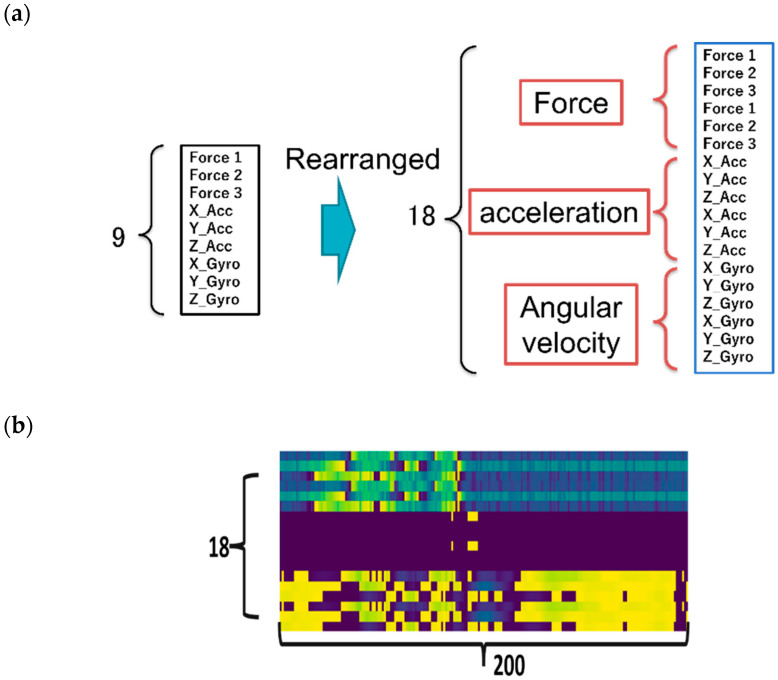
(**a**) The structure of a dataset. (**b**) the resultant dataset, virtual image.

**Figure 7 sensors-22-07840-f007:**
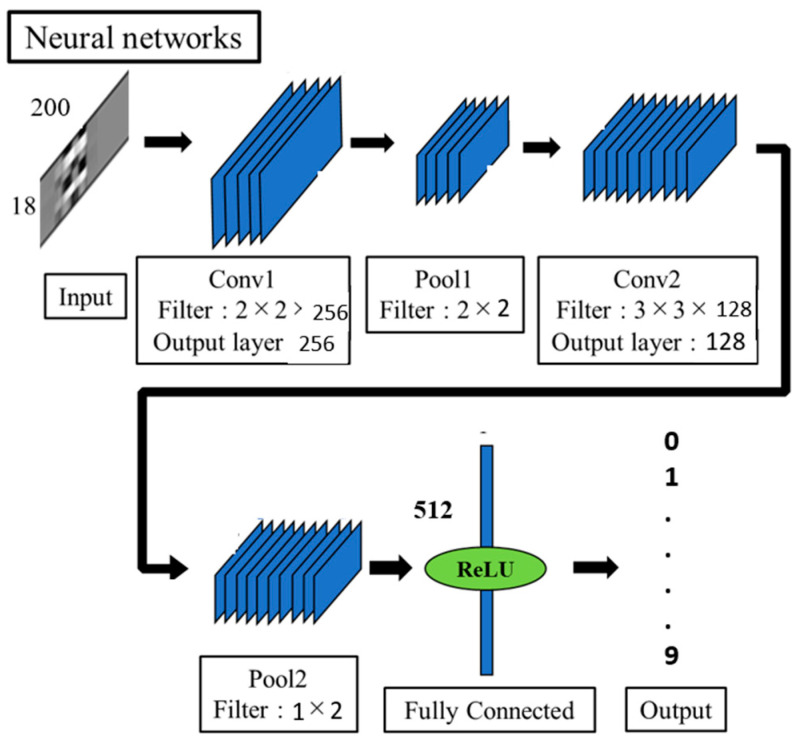
The architecture of CNN.

**Figure 8 sensors-22-07840-f008:**
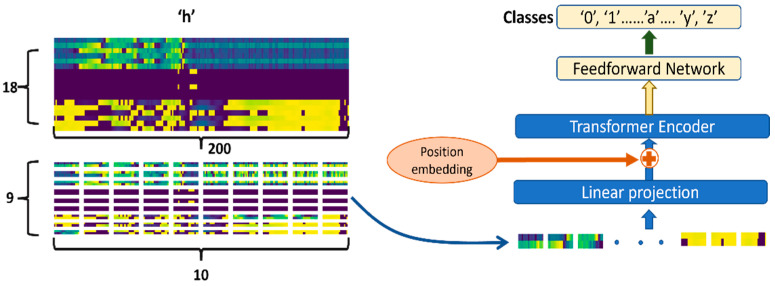
The structure of the transformer network, ViT.

**Figure 9 sensors-22-07840-f009:**
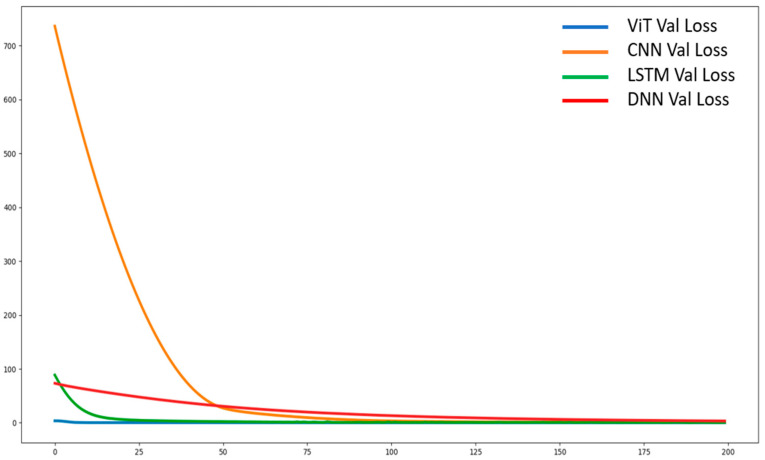
Loss of the validation datasets.

**Figure 10 sensors-22-07840-f010:**
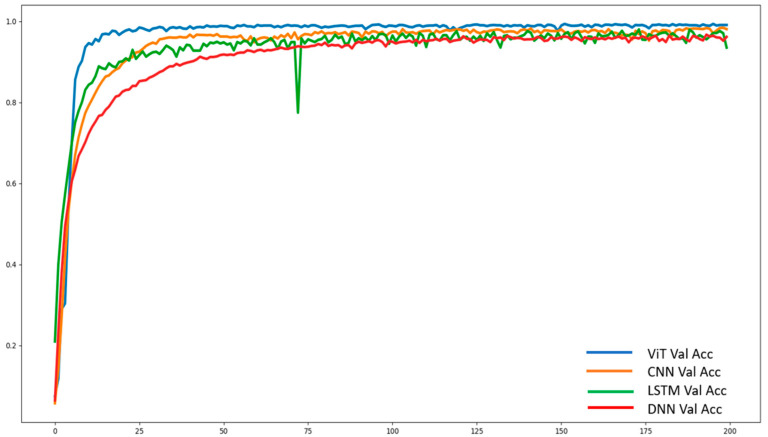
Accuracy graphs of the validation datasets.

**Figure 11 sensors-22-07840-f011:**
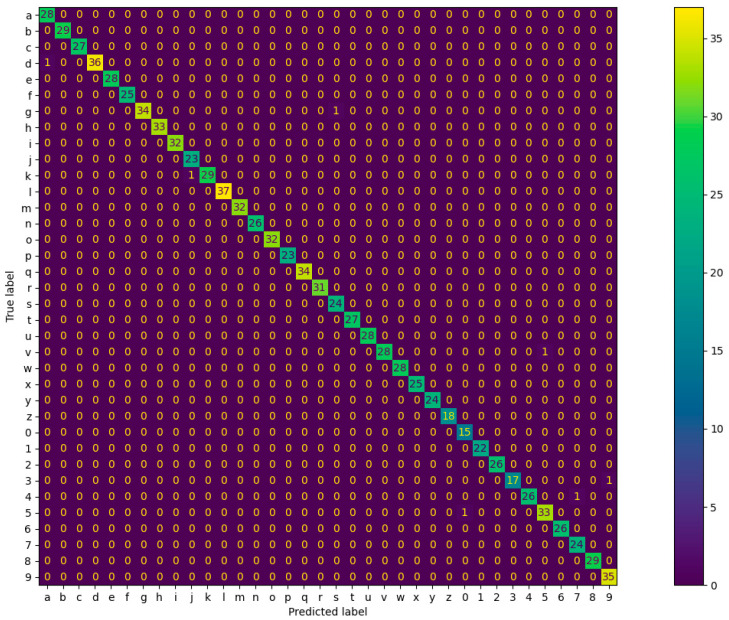
Confusion matrix for the testing datasets.

**Figure 12 sensors-22-07840-f012:**
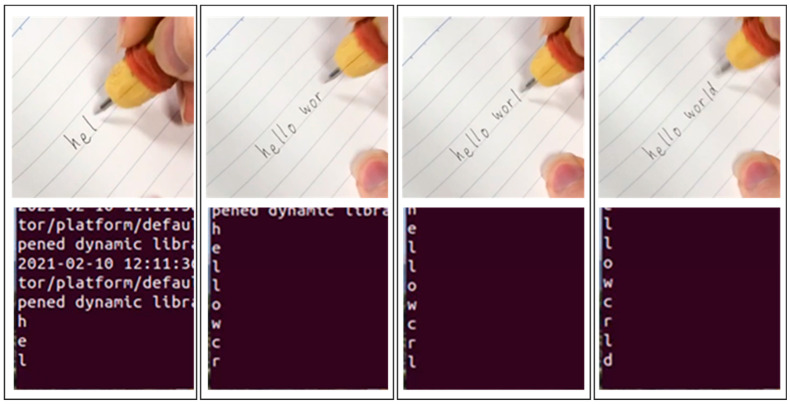
Real-time handwritten character recognition example.

**Table 1 sensors-22-07840-t001:** Specifications of LSM9DS1 IMU sensor.

Quantity	Value
3-axis acceleration	±4 [g] with resolution 0.122 [mg]
3-axis angular rate	±2000 [dps] with a resolution of 70 [mdps]
3-axis magnetic field	±400 [uT] with a resolution of 0.014 [uT]
Data output	16-bit
Serial interfaces	SPI/I^2^C
Power supply	1.9 [V] to 3.6 [V]

**Table 2 sensors-22-07840-t002:** Specifications of HSFPAR303A force sensor.

Quantity	Value
Dimension	4.0 × 2.7 × 2.06 [mm]
Force range	0–7 [N]
Sensitivity	3.7 [mV/N]
Supply Voltage	1.5 [V] to 3.6 [V]

**Table 3 sensors-22-07840-t003:** Validation accuracy results of the neural network models.

	Networks			
Data Type	ViT	CNN	LSTM	DNN
IMU only	97.82%	96.94%	95.78%	95.51%
Force only	98.20	96.28	73.76	82.58
IMU + Force	99.05%	97.89%	97.21%	95.65%

**Table 4 sensors-22-07840-t004:** Additional evaluation metrics for the ViT trained model.

Macro-Averaged Precision	Macro-Averaged Recall	Macro-Averaged F_1-Score_
0.9923	0.9932	0.9926

## Data Availability

Datasets used in this research can be found here. https://github.com/tsgtdss583/DigitalPen-Dataset.

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
