# Peer review of "Deep-Learning-Based Character Recognition from Handwriting Motion Data Captured Using IMU and Force Sensors"

_sensors, 2022, doi:10.3390/s22207840_

Round 1

Reviewer 1 Report

This manuscript employed deep learning technique for character recognition from handwriting motion data captured using IMU and force sensors, where four models, namely Vision transformer (ViT), DNN (Deep Neural Network), CNN (Convolutional Neural Network), and LSTM (Long Short-Term Memory) were developed. Finally, the experimental verification was conducted to prove the performance of the proposed models, with satisfactory results. Overall, the topic of this study is interesting, and the manuscript was well organised. I suggest that it can be accepted for publication in Sensors, if the authors can well address the following comments.

1.       The contribution and innovation of the manuscript should be clarified clearly in abstract and introduction.

2.       Broaden and update the literature review about application of deep neural networks or deep learning for resolving practical problems. E.g. Crack detection of concrete structures using deep convolutional neural networks optimized by enhanced chicken swarm algorithm; A novel deep learning-based method for damage identification of smart building structures.

3.       As is well known, once the architecture is determined, the performance of deep learning model is related to the hyperparameter setting of network. How did the authors adjust hyperparameters of the proposed models to achieve the optimal prediction performance?

4.       A parametric study is necessary to evaluate the robustness of the deep learning models for character recognition.

5.       Besides confusion matrix, are there any evaluation metrics for comprehensively comparing the performance of different deep learning models?

6.       More future research should be included in conclusion part.

Reviewer 2 Report

The paper presents a writing gesture recognition approach using deep learning with several different sensors. The paper is well presented however, i'd like to ask few questions

- What is the meaning of ofline methods in the abstract. isn't it too ealry to introduce the word without providing the defination in ragard to this study

- Other than catagorization of offline and online, there is another catagorization within the offline which is touch based and non touch based. Authors should mention that as well you may check the article 'Hand gestures recognition using radar sensors for human-computer-interaction: A review'

- I wonder why the authos have not included the Force sensor only sensor results. IMU only sensor results are included in table 3 but not the force only.

- Add more details and figures regarding the similar kind of writing gestures. for example, how the raw data looks like for the case of 'O' and 'Q' etc.

- How the i and l are separated i wonder similarly how zero and 'o' are separated?  

- Other than accuracy add the precision reacall f1 score as well. 

- Were they left handed or right handed participants? will it make any difference? what about the speed of gestures?

Round 2

Reviewer 2 Report

My concerns have been answered properly by the authors.